# A Novel Method for Achieving Precision and Reproducibility in a 1.8 GHz Radiofrequency Exposure System That Modulates Intracellular ROS as a Function of Signal Amplitude in Human Cell Cultures

**DOI:** 10.3390/bioengineering12030257

**Published:** 2025-03-04

**Authors:** Cyril Dahon, Blanche Aguida, Yoann Lebon, Pierre Le Guen, Art Dangremont, Olivier Meyer, Jean-Marie Citerne, Marootpong Pooam, Haider Raad, Thawatchai Thoradit, Nathalie Jourdan, Federico Bertagna, Margaret Ahmad

**Affiliations:** 1Laboratoire de Génie Electrique et Electronique de Paris, Sorbonne Université/CNRS, F-75005 Paris, Francepierre.le_guen.1@etu.sorbonne-universite.fr (P.L.G.);; 2Institut de Biologie Paris-Seine, Sorbonne Université/CNRS, F-75005 Paris, France; 3Institut Jean Le Rond d’Alembert, Sorbonne Université/CNRS, F-75005 Paris, France; 4Department of Biology, Faculty of Science, Naresuan University, Phitsanulok 65000, Thailand; marootpongp@nu.ac.th; 5Engineering Physics Program, Xavier University, Cincinnati, OH 45040, USA; raadh@xavier.edu; 6Institute for Quantum Biology, University of Surrey, Guildford GU2 7XH, Surrey, UK; 7Department of Biology, Xavier University, 3800 Victory Parkway, Cincinnati, OH 45207, USA

**Keywords:** radiofrequency fields, telecommunications, reactive oxygen species (ROS), oxidative stress, microwaves, human cell culture, hormesis

## Abstract

Radiofrequency fields in the 1–28 GHz range are ubiquitous in the modern world, giving rise to numerous studies of potential health risks such as cancer, neurological conditions, reproductive risks and electromagnetic hypersensitivity. However, results are inconsistent due to a lack of precision in exposure conditions and vastly differing experimental models, whereas measured RF effects are often indirect and occur over many hours or even days. Here, we present a simplified RF exposure protocol providing a single 1.8 GHz carrier frequency to human HEK293 cell monolayer cultures. A custom-built exposure box and antenna maintained in a fully shielded anechoic chamber emits discrete RF signals which can be precisely characterized and modelled. The chosen amplitudes are non-thermal and fall within the range of modern telecommunication devices. A critical feature of the protocol is that cell cultures are exposed to only a single, short (15 min) RF exposure period, followed by detection of immediate, rapid changes in gene expression. In this way, we show that modulation of genes implicated in oxidative stress and ROS signaling is among the earliest cellular responses to RF exposure. Moreover, these genes respond in complex ways to varying RF signal amplitudes consistent with a hormetic, receptor-driven biological mechanism. We conclude that induction of mild cellular stress and reactive oxygen species (ROS) is a primary response of human cells to RF signals, and that these responses occur at RF signal amplitudes within the range of normal telecommunications devices. We suggest that this method may help provide a guideline for greater reliability and reproducibility of research results between labs, and thereby help resolve existing controversy on underlying mechanisms and outcomes of RF exposure in the general population.

## 1. Introduction

Oscillating electromagnetic fields are ubiquitous in the modern world, emanating from telecommunications equipment, household appliances, and microwave sources. Indeed, billions of individuals worldwide are in daily contact with radiofrequency fields (RF) in the 1–28 GHz range through the use of mobile phones and personal electronic devices, stimulating significant research into the potential physiological effects of such exposure. There have been reported changes in gene expression, DNA repair, cognitive and behavioral modifications, cancer risks, neurological effects, and impacts on reproductive health and fertility [1].

Currently, RF exposure is generally considered safe. However, guidelines on RF exposure limits primarily address heat-related issues [2], whereas it is increasingly evident that significant physiological effects can occur at RF exposures much lower than the ones required to generate thermal effects [3,4]. Recent studies have demonstrated that a primary effect of RF emissions is the induction of mild cellular oxidative stress, as measured by increased production of reactive oxygen species (ROS) within minutes of exposure [5,6,7,8,9]. This finding is particularly promising as such modulation of cellular ROS could potentially explain many of the varied and complex physiological consequences that have been reported in response to RF exposure, including effects on gene expression, DNA repair, cognitive and behavioral changes, cancer risks, and impacts on reproduction and fertility [9]. Nonetheless, the literature is filled with conflicting reports and these conclusions remain controversial (see [10] and references therein).

As a consequence, resolving areas of controversy and obtaining results with suitable precision and reproducibility remain major obstacles to progress in the field. The electromagnetic systems for the exposure of biological samples must meet both biological and electromagnetic constraints [11]. Even for in vitro cell cultures, which in principle represent a homogeneous and reproducible biological system, the challenges are significant. The experimenter must ensure that the RF signal provides homogeneous cell exposure, and accurate measurement of the signal under the conditions of the experiment must be ensured. Contamination from unwanted electromagnetic field sources (such as cellular telecommunications and Wi-Fi), which are always present in the environment, must be minimized. Furthermore, the physical conditions necessary for cell culture growth have to be reliably provided, along with a sham experiment under virtually identical conditions.

Traditionally, controlled electromagnetic exposure systems for use with cell cultures include propagating exposure systems such as TEM (transverse electromagnetic) cells (e.g., [12,13,14]) and enhanced transmission line-based systems (e.g., [12,15,16]). These devices are placed inside a commercial incubator (e.g., [13,14,15,16,17,18]) or can be directly temperature-controlled (e.g., [12]). Certain devices also allow the simultaneous use of a microscope (e.g., [17]). The challenge for these systems has been maintaining appropriate incubation conditions around the cell cultures (temperature, CO_2_, and humidity) throughout relatively long experimental procedures; ensuring a suitably controlled sham condition; and achieving precision and homogeneity in the exposure signal. A further problem has been that contamination from external signals has not been consistently eliminated. An exception has been GTEM cells [14], which can function as a Faraday cage, or else the exposure systems can be placed in an anechoic chamber [19,20]. However, such devices are often complex and expensive to build and calibrate and have not been consistently used. Finally, most studies have been performed using relatively long (hours to days or weeks) periods of RF exposure. Thus, the measured consequence (e.g., cancer and behavioral changes) are only very indirectly related to the initial RF trigger. All of these factors have significantly contributed to the shortage of reliable results and the ongoing controversy in the field.

In the current study, we overcome many of the limitations of previous research by treating cell cultures with only a single 15 min exposure to 1.8 GHz carrier wave, which has been previously reported to induce rapid and significant changes in intracellular ROS (reactive oxygen species) [9]. Since only a short exposure period is provided during the entire experiment, both sham and RF-treated cells can be grown and maintained in the same incubator under identical growth conditions through the bulk of their growth cycles. Moreover, the cells are exposed to the single RF signal in a custom-built, separate, standalone exposure system that achieves the required maximum homogeneity of the electromagnetic field on the biological samples [19]. The device is small enough to be easily moved and placed in an anechoic chamber. It is simple to construct, operate, and to model numerically; therefore, it can be readily reproduced in other laboratories. Importantly, the system can be easily modified for the use of complex waveforms, such as those used in telecommunications, simply by changing the antenna. Because both exposed and sham samples are treated in parallel and in the identical way, an accurate baseline and control condition is obtained with no risk of contamination by the RF exposure treatment. In this manuscript we refer to our emission box as an RF exposure box.

As a proof of principle of this approach, we provide a dose-dependent assessment of cellular response in the signal amplitude range of common telecommunications devices. A single short 15 min exposure to a basic single-frequency sinusoidal carrier wave (1.8 GHz) was used to irradiate human cell cultures in the non-thermal power flow amplitude between 1 × 10^−2^ and 1 × 10^−7^ Wm^−2^. The modulation of ROS (reactive oxygen species)-regulated gene expression was used as biological readout.

## 2. Materials and Methods

### 2.1. Cell Cultures, Growth, and Exposure Conditions

Cell cultures and growth conditions were as previously described [20]. Briefly, human embryonic kidney (HEK) 293 cells were cultured in a CO_2_ incubator (MCO-18AC, Panasonic Biomedical, Leicestershire, UK), at 37 °C and 5% CO_2_. Cells were grown in 75 mL culture flasks containing 10 mL Modified Eagle medium (MEM; Sigma, St. Louis, MO, USA) and subcultured every 4 days. For the experimental assays, HEK-293 cells were seeded at a density of 2 × 10^6^ cells in 22.1 cm^2^ flasks and incubated for 24 h at 37 °C/5% CO_2_ to a final density of around 5 × 10^6^ cells. Both sham and test samples were diluted from the same parent cell culture stock to ensure consistency.

Exposure conditions involved placing the test and control samples in identical positions inside the RF exposure box for 15 min. The test samples were exposed to the RF signal, while the control (sham) samples underwent mock exposure without the RF signal. After the exposure, both sham and test cell cultures were returned to their regular incubator for a further 2 h 45 min to allow gene expression response to occur. Following this incubation, cell pellet samples were harvested into liquid nitrogen for gene expression analysis. Each condition was evaluated using three biological replicates on three independent days, starting from independently generated parent cell culture stocks.

### 2.2. Quantitative RT-PCR Analysis of Altered Gene Expression

Quantitative PCR (qPCR) analysis was performed as previously described [9]. After exposure to each treatment condition, total RNA was extracted from HEK293 cells using the Total RNA Miniprep Kit (New England Biolabs, Ipswich, MA, USA). cDNA was prepared from 1 µg total of RNA using the SuperScript first-strand synthesis system (Thermo Fisher Scientific, Waltham, MA, USA). Quantitative RT-PCR was performed using Luna qPCR master mix (New England Biolabs, Ipswich, MA, USA). Primers for RF exposure-regulated genes previously identified (K1AA, KRT79, DDX-50, RPS16P5, GPX1, and SOD2) were used as described in prior studies [9]. Primers used in this study are listed in Table 1. The GADPH gene was used as the reference gene. Quantitative real-time PCR was performed using Fast SYBR Green qPCR Master Mix (CFX Opus 384 Real-Time PCR System). The cycling conditions were as follows: an initial denaturation step at 95 °C for 1 min, followed by 40 cycles of 95 °C for 15 s, and annealing/extension at 60 °C for 45 s in CFX Opus 384 Real-Time PCR System (Bio-Rad, Hercules, CA, USA). Three biological replicates were performed for each gene (*n* = 3). Data analysis to determine the relative expression level of the target genes was performed as previously described [9]. Details of the primers used for gene expression analysis are provided in Table 1.

### 2.3. Statistical Analysis

All data were analyzed using GraphPad Prism version 7.4.2 for Mac (GraphPad Software, La Jolla, CA, USA). Normality was assessed using the Shapiro–Wilk test. Results are expressed as the mean ± standard error of the mean (SEM). To compare the differences between treated and control conditions for each gene, a one-way ANOVA was performed, followed by Tukey’s multiple comparison test. Comparisons were made between exposed samples (to a given RF condition) or Sham-exposed samples and control samples (less than 6.6 × 10^−7^ Wm^−2^), which were passaged at the same time from the same parent cell stock. Statistical significance was defined as follows: *p*-value < 0.05 (*), <0.01 (**), and <0.001 (***).

### 2.4. The RF Exposure System

The overall experimental setup is illustrated in Figure 1. The Petri dish containing cell cultures to be irradiated is placed in the RF exposure box, which in turn is housed within an anechoic chamber. This isolates the sample from any external electromagnetic radiation. To maintain this isolation, the control and measurement electronics are positioned outside the chamber.

The RF exposure box is opaque, blocking the passage of visible light. A PT100 temperature sensor installed inside the box, regulates the circulation of hot water through a PID servo-control. This ensures a stable internal temperature of 37 +/− 0.2 °C during the experiment.

The RF exposure treatment is delivered via a patch antenna, externally fed by a continuous wave generator, providing waveforms of known power and frequency to the antenna. A Vector Network Analyzer is used to characterize the antenna and its feed cables.

To ensure accuracy in the determination of the electromagnetic signal, an electric field probe was installed for direct measurement of the incident electric field at the position of the Petri dish. The received electrical power was measured with a spectrum analyzer and then converted into electromagnetic power values.

For the design and phenomenological characterization of the electromagnetic field within the RF radiation box, numerical simulations were performed using CST Studio Suite, 2019 version.

### 2.5. Detailed Technical Description of the RF Exposure Device

#### 2.5.1. The Geometric Aspects

Figure 2 illustrates the electromagnetic exposure box. The ideal conditions would involve a device operating in air or vacuum (“free space” condition) with infinite dimensions (“far-field” condition) in which a plane wave propagates. To estimate the distance between the microwave emitter and the observation point (i.e., between the patch antenna and the Petri dish) the “far field” must be satisfied. We therefore used the generally accepted formula: $d\geq \frac{2D^{2}}{\lambda }$ where d is the minimum distance, D is the largest dimension of the electromagnetic source (here, the diagonal of the patch antenna ~5.5 cm) and λ is the wavelength of the propagating wave. The wavelength associated with our working frequency of 1.8 GHz is of the order of 16.5 cm in air, i.e., d > 3.6 cm. The “free space” condition was approximated by choosing building materials that are as “transparent” as possible in the electromagnetic sense. This translates into materials with low real permittivity and even lower conductivity, in our case poly(methyl methacrylate) (PMMA). This has the characteristics of a very good electrical insulator.

Numerical simulations are essential for gaining a realistic understanding of the electromagnetic phenomena at work in this exposure system.

#### 2.5.2. The Thermal Aspects

A major concern with microwave irradiation protocols is the risk of artifacts due to heating of the samples. In our RF exposure box, the inside of the RF radiation box is designed to be maintained at a temperature of 37 +/− 0.2 °C with minimal heat loss.

Figure 2B shows a cross-section of the RF box design for simulation with CST. The box consists of a double shell made of poly(methyl methacrylate) (PMMA). Between these two shells, on the sides and inside the hood, we have placed extruded polystyrene (XPS) sheets to act as thermal insulation. Figure 2C shows the underside of the inner shell. A groove has been cut by a CNC milling cutter, following a serpentine path over the entire surface. This groove accommodates a silicone tube through which hot water circulates. The water heats the air inside by thermal conduction. A hole is drilled in a side wall to allow a PT100 temperature sensor and/or an electric field probe to pass through.

Outside the anechoic chamber, a computer controls an Arduino microcontroller. This receives data from the MAX31865 temperature sensor connected to a 3-wire PT100. The conversion of the PT100 resistance to temperature was calibrated using a precision-calibrated ETI Thermapen Thermometer. The calibration protocol followed is described in https://www.analog.com/media/en/technical-documentation/application-notes/AN709_0.pdf (accessed on 19 February 2025).

The measured temperature serves as input to a PID control algorithm programmed in the microcontroller (https://github.com/br3ttb/Arduino-PID-Library/ (accessed on 19 February 2025)), whose characteristic time constants have been obtained from the system’s step response and then transformed into PID coefficients according to Internal Model Control rules. The PID servo output drives the duty cycle of a 5 s period PWM signal.

This signal, via a power driver, switches a pump on or off in a beaker of water heated to 60 °C. Hot water is then sent to the incubator as needed to maintain an internal temperature of 37 +/− 0.2 °C. The computer (Figure 1) records temperature data in real time during the experiment. Figure 2D shows an entire PID temperature control cycle, starting from background temperature before the beginning of the experiment (at t = 0). The initial heating phase first brings the box to 37 °C prior to sample placement. This is followed by the two hood openings representing the insertion and subsequent removal of the Petri dish. When the hood is opened, there is a small (less than 1 °C) transient drop in temperature which the PID controller restores to the set point of 37 °C (Figure 2D, hood openings). However, at no time does the temperature exceed the system’s temperature limits. Thus, we can rule out the main concern of heating artifacts caused by the RF exposure setup.

#### 2.5.3. The Electromagnetic Aspects

Figure 3 shows the microwave emission system inside the RF box. Outside the anechoic chamber, a microwave generator (Anritsu MG3692B) feeds a patch antenna (double-sided PCB, Rogers corp., RO3006) via an SMA cable (see Figure 1 for overview of setup). This cable first passes through the wall of the anechoic chamber via a dedicated passage, then crosses the RF box cover and the layer of thermal insulation. It feeds the patch antenna, which then radiates downwards.

#### 2.5.4. The Simulated Antenna

The antenna was designed and optimized using CST electromagnetic simulation software, to operate at 1.8 GHz. The signal we wish to transmit is a pure sinusoid, the most elementary waveform. Appendix A shows the CST-optimized design and the S11 reflection coefficient. This indicates a minimum reflection of −14 dB between the 50 Ω impedance input port and the patch antenna, resulting in maximum radiation at 1.8 GHz. It has been designed to radiate a linear polarization according to the u axis. This antenna has a relatively narrow bandwidth, making it suitable for transmitting a pure sinusoidal signal, but not for modulated signals, as in telecommunications. Nevertheless, it could potentially also be used to transmit a pulsed field (sinusoid at 1.8 GHz amplitude-modulated by a square-wave signal) but with a low repetition frequency.

#### 2.5.5. The Actual Antenna

Appendix A is to be compared with Appendix A. It shows the actual antenna (left) engraved from the CST design using a Protolaser PLS4 laser engraver from LPKF Laser & Electronics Tualatin, OR United States. On the right is a photo of the S11 reflection coefficient measured with a Vector Network Analyzer (VNA HP 8720D from Agilent Technologies, Lexington, MA, USA). The actual optimum frequency of this antenna is 1.77 GHz instead of 1.8 GHz simulated frequency. This actual frequency is used for the experiment.

#### 2.5.6. The RF Exposure System (Simulations)

Appendix A shows the design of the exposure system inside the box. The antenna is placed in the center of the (u, v, w) reference frame. The polystyrene Petri dish (in red) is held at a fixed distance from the antenna by a custom-made 3D printed Petri holder made of polylactic acid (PLA) (in white). The antenna radiates downwards, in the positive direction of the w vector. A 6 mm-thick mat of RGD-S-124 absorbent material from Cuming Microwave (in green) is placed to damp any reflections on the bottom of the inner shell and minimize any residual cavity effects. The simulation parameters are given in Appendix A.

On the right side of Appendix A, the distance of 170 mm between the antenna and the outer bottom surface of the Petri dish is shown. The Petri dish is 1 mm thick, and an additional distance of 1 mm is added above the inner bottom surface of the Petri dish as the field measurement plane, giving a total distance of 168 mm.

Appendix A shows the results of the CST simulations in the XY plane, at a distance of 168 mm from the patch antenna. This means that the observation plane is inside the Petri dish, where the cell culture should be if simulated.

The electrical power “sent” to the antenna port by CST is its calculation standard, i.e., 0.5 W. This does not correspond to the real powers we have used. The orders of magnitude of the simulations can in no way be compared with our measurements. Simulations are then useful for phenomenological study, but in no way predictive.

Appendix A on the left gives the electromagnetic power (Wm^−2^) in the plane of interest. This power is the Poynting vector, the vector product of the electric $\vec{E}$ and magnetic $\vec{H}$ fields, calculated by the CST simulator. The red circle, ~52 mm in diameter, represents the edge of the Petri dish. Each pixel measures 1 mm^2^. This simulation shows the power inhomogeneity within the red circle. We have decided to quantify this by the percentage of average power represented by the standard deviation of power. This is:%inhom.=σ(P(x,y))P(x,y)¯*100∣(x,y)∈Petri≃7%

Appendix A on the right shows the CST simulation of the electric field vectors $\vec{E}(x,y)$ in the Z = 168 mm plane. This shows the polarization of the wave. This is predominantly linear, oriented along the $\hat{x}>0$ axis. The polarization ratios are such that: EZ¯EX¯=0.053 and Ey¯EX¯=0.015.

The weakness of the $\bar{\left|E_{Z}\right|}$ component also indicates that the wave is beginning to have a locally plane-wave structure. This supports the validity of the far-field hypothesis at the chosen distance of 168 mm. But the simulation provides an additional virtual experiment. If the far-field and therefore locally plane-wave hypothesis is valid at this distance, the calculation of the mean power by the Poynting vector can also be performed using the electric field alone, such as Ppoynting=ℜ(12E→∧H*→)=E22η=Pplanewave, where $\vec{E}$ and $\vec{H}$ are the complex expressions of the electric and magnetic fields, * represents the conjugation operator in the space of complex numbers and η is the wave impedance in the propagation medium. The simulation results, for a distance of 168 mm, give:PplanewavePpoynting≃1.01

This simulation result indicates that the far-field, locally plane-wave approximation is acceptable. Electromagnetic power can be deduced solely from the electric field, without needing to reference the magnetic field. This allows the use of an electric field probe alone to obtain a valid estimate of electromagnetic power. This is the technique chosen for power measurements.

#### 2.5.7. RF Exposure of the Petri Dish and the Probe Holders (Measurements)

The simulated Petri dish holder in Appendix A is shown holding a Petri dish. The Petri holder’s modular design allows the Petri dish and the four arms supporting it to be removed to install a bracket carrying an electric field probe, Appendix A. The jib system is dimensioned so that the center of the dipole, which senses the electric fields, is positioned at a distance of 168 mm from the patch antenna when the probe holder is placed in the RF radiation box. The Enprobe’s EFS-105 electric field probe consists of a head (Appendix A) containing the electrodes of the dipole that senses the electric field. The probe head is the only part containing metal, while the cable is an optical fiber. The small size of the probe and its materials limit the field modification implied by its presence. The electric field picked up by the dipole is converted into an optical signal, which is transmitted through the optical fiber to an electronic device (Appendix A), where it is reconverted into an electrical signal. An SMA cable connects this box to an ANRITSU MS2711D spectrum analyzer, which measures electrical power. The transformation of this electrical power, measured in Watts, into electromagnetic power in Wm^−2^ is carried out using an Antenna Factor obtained from the manufacturer during calibration of the probe. However, since the probe is a dipole, it can only measure an electric field in one direction at a time, along the x, y or z axis (respectively u, v, w axes for CST simulations). Therefore, the probe holder can be manually rotated to measure power components P_EMG_^X^, P_EMG_^Y^, and P_EMG_^Z^ in Wm^−2^.

This probe is also used for indirect measurement (see Section 2.5.8 and Appendix A).

#### 2.5.8. Preliminary Electromagnetic Measurements

The RF box is heated to 37 °C with water circulating at 60 °C. The HF generator is connected to the patch antenna via its SMA cable. The chosen frequency is 1.77 GHz. This is the resonant frequency of the actual antenna (Appendix A). The electric field probe is installed in the RF radiation box, in place of the Petri dish. The probe is connected to its housing by the optical fiber. The housing is then connected to the spectrum analyzer via a second SMA cable.

A first experiment serves as a phenomenological check against the simulation. The generator’s transmit power is set to +10 dBm. The field probe is oriented to successively measure the amplitude of the electric field along the three axes (x, y and z). Appendix A is a photo of the electric power measurement on the x axis, i.e., P_EMG_^X^ = 3.2 × 10^−3^ Wm^−2^ @ 1771.17 MHz. On the other axes, we measured P_EMG_^Y^ = 5.3 × 10^−6^ Wm^−2^ and P_EMG_^Z^ = 4.2 × 10^−5^ Wm^−2^. We observed the behavior indicated by the simulation (Section 2.5.6). The electric field is predominantly oriented along the x axis (u axis for CST) and very little along the other two axes, validating the locally plane-wave approximation. The photo in Appendix A also shows that the spectrum analyzer cannot measure powers below P_MES_ = −75 dBm corresponding to P_EMG_ = 3.3 × 10^−7^ Wm^−2^, which is the sensitivity limit of the device.

The aim of the second experiment is to match the electrical powers measured by the probe to the electrical powers emitted by the HF generator. The aim of this operation is to give an order of magnitude of the power incident on each biological sample.

Appendix A shows a photo of the S12 transmission coefficient measurement of the SMA cable between the probe housing and the spectrum analyzer. This measurement is made by the VNA over a frequency range from 1 GHz to 3 GHz. Its attenuation is −3.896 dB at 1.77 GHz. The electrical power measured by the spectrum analyzer is then transformed into electromagnetic power using the following formula:PELEC=PMESACABLEUELEC=PELEC⋅Z0E=AF⋅UELECPEMG=E2η0
where $P_{MES}$ is the power in Watts measured by the spectrum analyzer. $A_{CABLE}$ = 4.0776 × 10^−1^ is the attenuation coefficient of the cable. $P_{MES}$ must be divided by $A_{CABLE}$ to compensate for the attenuation that the measured power undergoes as it passes through the cable. $P_{ELEC}$ is therefore the electrical power in Watts that is measured just outside the field probe housing. This power is used to calculate an RMS voltage in Volts, $U_{ELEC}$ thanks to $Z_{0}=50\Omega$, the impedance of the SMA connection port on the housing. The amplitude of the electric field E is then calculated using the antenna factor given by the manufacturer for a frequency of 1.77 GHz, $AF=179.89\;m^{-1}$. The electromagnetic power of the wave $P_{EMG}$ is deduced from the amplitude of the electric field using $\eta_{0}≃377\Omega$, which is the wave impedance of the air. This last formula is only valid for a locally plane wave, an assumption we have verified.

Note: the change from power in dBm to power in Watt is made by:$$P_{WATT}=10^{\left(P_{dBm}/10\right)}/1000$$

Table 2 shows the results obtained for several power levels $P_{gene}$ emitted by the generator. Measurements of $P_{MES}$ are made on the spectrum analyzer and transformed into electromagnetic power $P_{EMG}$. This power represents the incident wave, at the Petri dish position, 1 mm above the bottom. Table 2 shows only the x-axis component of the power; the other components are negligible.

Two observations can be made: firstly, when the generator sends out a power of −40 dBm or less, the measurement device exceeds its capabilities. Secondly, the system as a whole is linear: an increase in the generator’s electrical power results in the same increase in the measured electrical power. Thus, we find the expected physics.

Finally, to estimate the order of magnitude of the heat transfer induced by the electromagnetic powers involved, we carried out a quick calculation:

If a Petri dish of diameter ϕ≃0.053 m (surface area of the Petri dish SurfPETRI=πϕ24) receives an incident wave of power $P_{EMG}=1.1\times 10^{-2}$ W.m^−2^ for Δt=0.25 h and this Petri dish contains a layer of water 2 mm thick (i.e., $V_{water}≃4.5\times 10^{-6}$ m^3^), considering that this system is lossless, i.e., that any calories supplied are accumulated to increase the temperature, the total rise in temperature is approximately 1.2 × 10^−3^ °C. This result is obtained by the formula:$$\Delta T=\frac{P_{EMG}\cdot Surf_{PETRI}\cdot \Delta t}{C_{water}\cdot V_{water}}$$

If the Petri dish is full of water with the water heat capacity $C_{water}=1160\;{\mathrm{Wh}/(m}^{3}K)$, the global temperature elevation is $\Delta T=1.2e^{-3}\;{}^{\circ}C$. If the Petri dish is full of kidney cells with a typical kidney heat capacity $C_{kidney}=3644J/(\text{L.K})={1012\;\mathrm{Wh}/(m}^{3}K)$, $\Delta T=1.37e^{-3}\;{}^{\circ}C$.

Assuming that the cell culture sample can be assimilated to water or kidney cells, the existence of an overall “microwave oven” type thermal effect seems to be irrelevant to our experiment.

## 3. Results

### 3.1. Experimental Design for RF Exposure

The aim of this experiment was to obtain accurate and reproducible information on how RF signal amplitude modulates cellular gene expression in human cell cultures. The experimental outline is shown in Figure 4.

All test and control sample repeats used for a given experiment were subcultured from the identical parent stock as described (methods). On the day of the experiment, test and sham plates were chosen at random from the growth incubator and transferred individually to the RF exposure box in the anechoic chamber (methods) for 15 min. They were subsequently returned to the growth incubator for 2 h 45 for gene expression response to occur, followed by qPCR transcript analysis (Figure 4, see also methods). The RF exposure box and the incubators were within 5 m distance of each other and the transfer time was negligible. Three or more test samples were exposed one at a time to RF in the RF exposure box, whereas the control samples were placed one by one in the RF box for the same amount of time (15 min) without activating the HF generator output. These are the sham samples which are used as controls (not exposed to RF). All other treatments and manipulations of sham and test sample plates were identical.

The temperature was a critical feature that was monitored throughout the course of each experiment and was maintained by the PID controller at a steady 37 +/− 0.2 °C (Appendix A) except for a transient drop of less than 1 °C during placement of the Petri dish (hood opening). At no time was there a measurable heating effect or temperature increase.

To monitor the RF exposure of the sample, we placed the electric field probe (Appendix A) in the anechoic chamber next to the RF exposure box. The probe senses the electric field in the anechoic chamber, oriented along the x-axis. This is an indirect measurement and in no way modifies the power received by the Petri dishes. When these measurements were possible, i.e., when there is sufficient power from the generator, they enable us to observe that the antenna inside the RF box is working and to gain an overview of the electromagnetic environment in the anechoic chamber between 1 GHz and 3 GHz, in a single polarization. Although it is only an approximation, Appendix A nonetheless shows that a single $\hat{x}$ polarized electric field is measurable, at a frequency of 1.77 GHz in complete agreement with our input signal.

### 3.2. Signal Amplitude-Dependent Gene Expression Is Triggered by RF Exposure

A number of genes have been identified that respond rapidly and specifically to transient changes in cellular ROS, particularly as induced by applied static and pulsed electromagnetic fields [19,20]. Since the primary mechanism of response to RF exposure also apparently involves the stimulation of intracellular ROS, we have accordingly monitored the expression of a number of genes linked to oxidative stress, and which moreover reportedly respond to the RF signal amplitude.

Expression characteristics of one of these genes (KIAA) are reported in Figure 5.

RF exposure resulted in a 2.5-fold stimulation in gene expression at P_gene_ = +10 dBm (corresponding to P_EMG_ = 2.9 × 10^−3^ Wm^−2^, the highest amplitude used). There was a progressive decline in gene expression levels at lower RF exposure amplitudes of P_gene_ = 0, −10, and −20 dBm (resp. P_EMG_ = 2.9 × 10^−4^, 2.9 × 10^−5^, 2.9 × 10^−6^ Wm^−2^), essentially identical to the control condition. However, at P_gene_ = −30 dBm (P_EMG_ = 6.6 × 10^−7^ Wm^−2^) and again at P_gene_ = −40 dBm there was a dramatic 5-fold increase in gene expression at these vanishingly low levels of RF amplitude (see Table 2 for power conversion). What is particularly striking is the biphasic, U-shaped nature of the response characteristic. These experiments were then repeated using five additional ROS responsive and/or ROS signaling genes (KRT79, DDX-50, RPS, GPX-1 and SOD2).

The results showed significant modulation of gene expression in response to RF exposure for the majority of the genes analyzed (Figure 6); KRT79 for example showed an almost 10-fold stimulation at P_gene_ = −10 dBm (P_EMG_ = 2.9 × 10^−5^ Wm^−2^), whereas at P_gene_ = 0 dBm (P_EMG_ = 2.9 × 10^−4^ Wm^−2^), there is actually a decrease in expression, by more than two fold. In fact, in all cases, the amplitude of gene expression was non-linear with respect to RF signal amplitude and showed mostly biphasic or U-shaped characteristics. For example, stimulation of KIAA was highest at P_gene_ = +10 dBm (P_EMG_ = 2.9 × 10^−3^ Wm^−2^) and below P_gene_ = −30 dBm (P_EMG_ = 6.6 × 10^−7^ Wm^−2^) but remained unresponsive in the intervening amplitudes (P_gene_ = 0, −10 and −20 dBm, resp. P_EMG_ = 2.9 × 10^−4^, 2.9 × 10^−5^, 2.9 × 10^−6^ Wm^−2^). Similarly maximal GPX1, SOD2 and RPS expression occurred in a relatively discrete range but was unresponsive to RF signal amplitudes either above or below this range. Such non-linear, U-shaped response patterns, which are consistent with prior results using completely different antennae and RF exposure conditions [9], are indicative of a nuanced biological response to exceedingly low signal amplitudes likely due to a receptor-mediated, hormetic cellular mechanism.

## 4. Discussion

Our exposure device provides a number of important features that ensure reliable and reproducible RF exposure conditions and experimental observations. Firstly, absence of contaminating radiation is ensured by the anechoic chamber, which in our case was from the GeePs laboratory (Methods), a state-of-the art facility used for antenna characterization. The use of an anechoic chamber greatly reduces any possibility of unknown contaminating signals, which could be extremely hard to detect at the low amplitudes that can cause biological effects. Secondly, a combination of numerical simulation and direct measurements greatly increase the level of precision in the exposure conditions. This is ensured by the fully characterized nature of the electromagnetic field exposure box. Thirdly, unlike the majority of RF exposure protocols, our study uses a homogeneous cell culture system exposed to only a single, short, simple carrier wave frequency. This ensures that our physiological readout (gene expression) is directly related to cellular responses triggered by the RF signal. This system thereby escapes from the uncertainty introduced by measuring indirect effects of RF exposure (such as the onset of cancer after several months), which are modulated in uncontrollable ways by many other physiological and environmental inputs. In sum, the improvements of this setup ensure a relatively low cost means to achieve results that are well characterized, reliable, and free from experimental artifact.

### 4.1. Cellular Response to RF Does Not Involve Thermal Effects

One of the key concerns in experiments involving exposure to GHz radiation is whether physiological effects can be quite simply due to increase in temperature, and therefore may result from artifact. Our exposure conditions are specifically designed to address this question by being conducted at extremely low power levels (see Table 2), such that a possible thermal effect of the “microwave oven” type can be ruled out. Thermal variation during the exposure was moreover measured and logged and there was at no point an increase in temperature throughout the course of the experiment (See methods in Appendix A). While indeed a minor (less than 1 °C) transient change in temperature occurs, this represents an actual decrease in temperature, occurring during lid opening of the box for insertion of the Petri plates and had no effect on cellular response. At no time was there an increase in temperature.

### 4.2. Human Cells Show a Physiological Response to RF Exposure at Amplitudes Below Current Safety Standards

An important biological finding of this study is that living cells show sensitivity to RF electromagnetic exposure at amplitudes (signal strength) that are far below current exposure safety limits. These effects furthermore occur well within the emission range of cellular phones and Wi-Fi antennae to which we are exposed daily. For instance, regular cellular phones can function anywhere within the range from +30 to −50 dBm (corresponding to 7.9 × 10^−2^ to 7.9 × 10^−10^ Wm^−2^ at a distance of 1 m in the simplistic case of an isotropic antenna and a perfect impedance adaptation), whereas a Wi-Fi router’s antenna emits at +20 to −30 dBm (7.9 × 10^−3^ to 7.9 × 10^−8^ Wm^−2^ in the same conditions). These values lie well within the range of the RF signal amplitudes used in this study (P_EMG_ = 2.9 × 10^−3^ to less than 6.6 × 10^−7^ Wm^−2^), and which in fact elicited up to 10-fold increase in expression of certain genes (Figure 6).

We compare the RF signal amplitudes used in our study with current safety standards as follows: our measurements of power flow emitted at the position of the cell cultures (Table 2) in our RF exposure box are 11 mWm^−2^ to 0.66 μWm^−2^ emission.

The most restrictive current safety conditions, taken from Table 5 p. 495 of the IRCNIP guidelines 2020 [21] (General Public exposure >400–2000 MHz averaged over 30 min and the whole body), indicate the following maximum thresholds:EMAX=1.375*f_MHz^0.5=58.336 Vm−1, HMAX=0.0037*f_MHz^0.5=0.15698 Am−1
$$P_{EMG}=f\text{\_}\mathrm{MHz}/200=9\;{\mathrm{Wm}}^{-2}\mathrm{with}\;f\text{\_}MHz=1800\mathrm{MHz}$$

These safety standards are orders of magnitude higher than the amplitudes used in the present study, and were largely arrived at on thermal considerations which are not relevant at these energies (see Table 2 and above Section 4.1).

To avoid any confusion as to the implication of our results for human health, we emphasize that our data cannot be taken as evidence that RF signals in this low range are dangerous. Indeed there is currently no conclusive evidence implicating RF exposure to disease of any sort. Furthermore, many environmental stressors including pollution, cigarette smoke, and strong sunlight induce an increase in intracellular ROS in human cells in the course of normal daily life. All these ‘normal’ sources of stress, including likely effects of daily RF exposure, are counteracted by powerful physiological antioxidant mechanisms that restore redox equilibrium and downregulate oxidative stress in healthy human cells (see below).

Nonetheless, it is apparent that RF signals, even of such astonishingly weak amplitudes, are not physiologically inert. Therefore, the primary physical mechanism of how these RF signals interact with cellular receptors, as well the direct physiological consequences on ROS stimulation and signaling pathways, represent important new branches of knowledge to be addressed in future.

### 4.3. RF Exposure Is Linked to Intracellular ROS and ROS Signaling Pathways

Our work is furthermore in basic agreement and supports numerous prior studies that have linked RF exposure to the induction of intracellular ROS (reactive oxygen species) and/or ROS signaling pathways [5,6,7,8,9]. This is a potentially significant mechanism as ROS has multiple physiological roles in cellular stress response pathways. For example, high levels of intracellular ROS can result in redox dysfunction, leading to oxidative stress which induces damage to lipids, proteins and DNA. Such damage has been implicated in pathological conditions such as ageing, diabetes, atherosclerosis, degenerative diseases and cancer [22], consistent with possible negative effects of RF exposure that have been documented in the literature. On the other hand, a milder induction of intracellular ROS can actually have a beneficial effect by triggering cellular adaptive and repair responses against oxidative damage. This characteristic of cellular response to ROS, has the consequence that the effects of RF exposure on cells could in fact be completely opposite depending on signal parameters, dose, exposure conditions and physiological readout [22].

Non-linear response characteristics and adaptation to the intensity of external stressors is a further feature of ROS signaling pathways. Due to the tightly regulated nature of redox and ROS homeostasis within cells, they can adapt to chronic increases in ROS by inducing ROS scavenging (destruction) pathways. As a consequence, the physiological effects of RF exposure are likely to be non-linear (i.e., not proportional to the RF input signal), as the higher the stimulation, the stronger the cellular antioxidant response and, thereby, the counterintuitive observation of actually lowered levels of intracellular ROS [23]. Furthermore long-term assays (continuous RF exposure over days or weeks) may not show responsivity, due to the cellular redox adaptation response. Therefore, transient effects of RF exposure may be a better means to detect physiological effects of RF at low signal amplitudes, and may have been missed in longer-term experimental setups.

### 4.4. Importance of Hormesis in Determining RF Safety Limits and Toxicology

Hormesis is a dose–response phenomenon characterized by a biphasic relationship, where low doses of a substance elicit a stimulatory or beneficial effect, while higher doses result in inhibitory or toxic outcomes [24,25]. For instance, compounds such as resveratrol, sulforaphane, and certain phytoestrogens have demonstrated biphasic dose–responses. At low concentrations, these substances may promote cellular proliferation or exert protective effects; at intermediate concentrations these substances show no physiological effects; while at higher concentrations they can inhibit cell growth or induce cytotoxicity [23,24,25].

Hormesis is particularly well documented for reactive oxygen species (ROS) signaling, when low levels of ROS may stimulate adaptive and beneficial cellular responses, while excessive ROS levels cause oxidative stress and cellular damage [22,23,26]. Examples include NRF2 pathway activation, where low ROS levels activate the nuclear factor erythroid 2-related factor 2 (NRF2) pathway, leading to the transcription of antioxidant enzymes such as superoxide dismutase (SOD), catalase, and glutathione peroxidase. This enhances cellular defense mechanisms [26,27]. High ROS levels, to the contrary, overwhelm antioxidant defenses, resulting in oxidative stress, mitochondrial dysfunction, and induction of apoptosis. Likewise in the case of mitochondrial hormesis (Mitohormesis), low ROS levels induce an adaptive response that enhances mitochondrial function and increases longevity, whereas high ROS levels cause mitochondrial damage, reduced ATP production, and cell death [27,28].

Mechanistically, hormesis is generally mediated by interaction with receptor-driven processes involving complex feedback loops, desensitization, and multiphasic interactions in downstream signaling pathways. For example, these can occur in the course of GPCR desensitization, stress adaptation pathways (like NRF2), cytokine receptor signaling and inflammation, and ion channel modulation [22,24,25]. Indeed, the present study has identified modulation by RF of genes for ROS scavenging enzymes such as superoxide dismutase (SOD) and glutathione peroxidase (GPX-1). These are directly involved in maintaining cellular redox homeostasis, and are known to be regulated by changes in the concentration of intracellular ROS as well as being subject to considerable feedback regulation through multiple enzymatic pathways. The biphasic response characteristics to RF are therefore an indication that RF likely interacts with biological receptors, possibly enzymes catalyzing redox reactions that give rise to ROS. Further experiments with cellular fractions and isolated enzymes may shed light on the underlying biochemical mechanisms involved.

Finally, the biphasic RF response characteristics are of direct relevance for the determination of risk, and underscore the importance of dose consideration in assessments of physiological effects of electromagnetic fields. Traditional models rely on the identification of a no observed adverse effect level (NOAEL) and assume that all doses below this threshold are without effect (see, e.g., [23]). However, the evidence for hormetic responses demonstrates that even if at certain amplitudes RF exposure induces no measurable effects, this does not in any way exclude that lower-dose exposures can have physiologically adverse effects [25,28]. To the contrary, our results are fully in keeping with a hormetic response mechanism in which signal amplitude is critical for evaluation of physiological effects. Therefore, a reevaluation of current RF exposure assessment practices is called for in order to account for the potential for low-dose stimulation and non-linear nature of dose–response relationships [28].

## 5. Concluding Remarks and Future Perspectives

We here describe an experimental RF exposure device and protocol that present fully characterized, defined RF signals to human cells in culture. Their effects are consistent with a biological receptor-driven mechanism whereby RF exposure modulates intracellular ROS and ROS signaling pathways. This provides a testable hypothesis for the many and varied effects of RF described in the literature.

These cellular responses occur at RF signal amplitudes that are orders of magnitude below those needed to achieve thermal effects, and lie within the signal range of personal electronic devices and mobile phones. Because this human cell response to RF is not linear as a function of the RF signal amplitude, the relation between RF exposure conditions and a physiological outcome is not readily deducible; indeed a robust gene expression response may occur at one amplitude but be undetected at another signal amplitude, or even undergo the opposite response entirely (e.g., opposite expression of the same gene at different signal amplitudes). It is therefore necessary to assess physiological response to RF signal exposure at multiple signal amplitudes and wavelengths, and preferably by using a readout assay that is rapid and direct. This may help explain existing confusion and contradictions in the literature, as well as stimulate future studies on the nature of the biological reception mechanisms.

Finally, although RF exposure from cell phones and telecommunications devices has not been proven harmful in any way, there is a definite physiological response in human beings to this signal range. Risk factors may therefore exist for susceptibility to RF exposure, for instance in individuals with reduced tolerance to oxidative stress and/or who are exposed to excessive stressors in their daily life. These additive or synergistic effects may contribute to certain poorly defined syndromes such as electromagnetic hypersensitivity (EHS) that have been linked to RF exposure in rare individuals in the past [29].

## Figures and Tables

**Figure 1 bioengineering-12-00257-f001:**
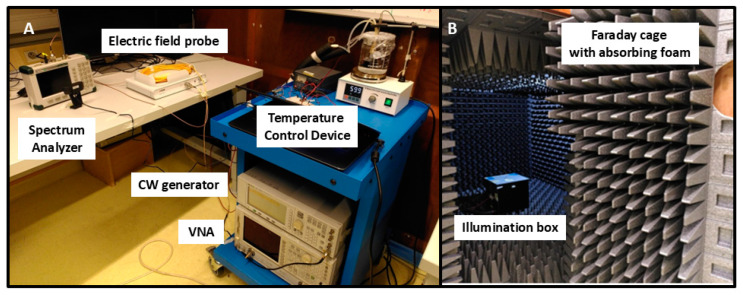
Overview of the electromagnetic exposure system. The RF exposure box is placed within the anechoic chamber (**B**) while the electric field probe, spectrum analyzer, temperature control device, and function generator are located outside the chamber (**A**).

**Figure 2 bioengineering-12-00257-f002:**
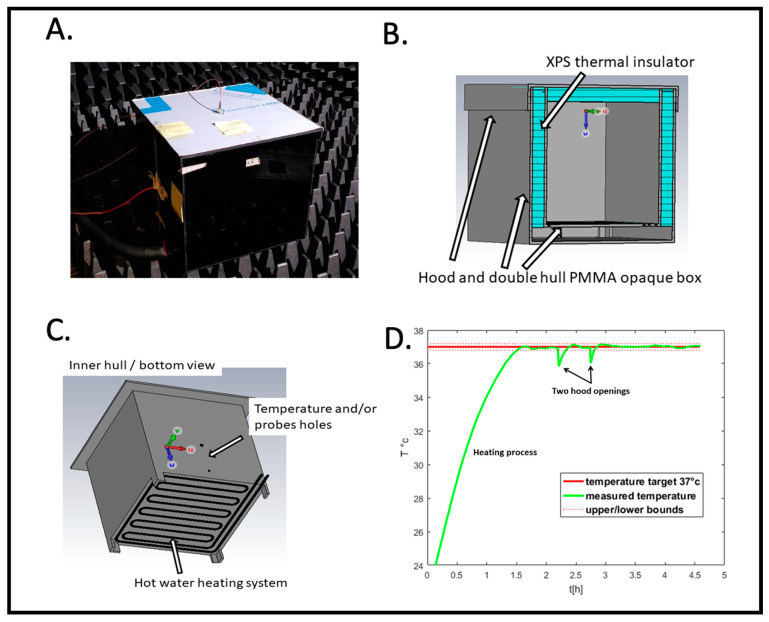
RF radiation exposure box. (**A**) Outside view placed in the anechoic chamber. (**B**) Structural features for thermal insulation. (**C**) Heating system. (**D**) Recording of the temperature as a function of time (hours) before and during the experimental cycle; example of PID controlled temperature. Samples are added and removed at the indicated ‘hood opening time’.

**Figure 3 bioengineering-12-00257-f003:**
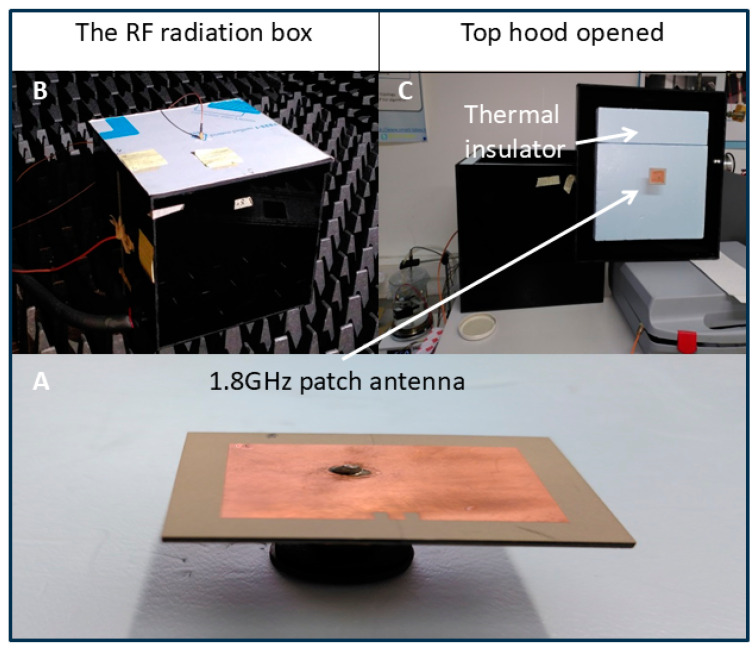
The patch antenna and its positioning. (**A**) The 1.8GHz patch antenna. (**B**) The RF radiation box. (**C**) Top hood opened shows the thermal insulator.

**Figure 4 bioengineering-12-00257-f004:**
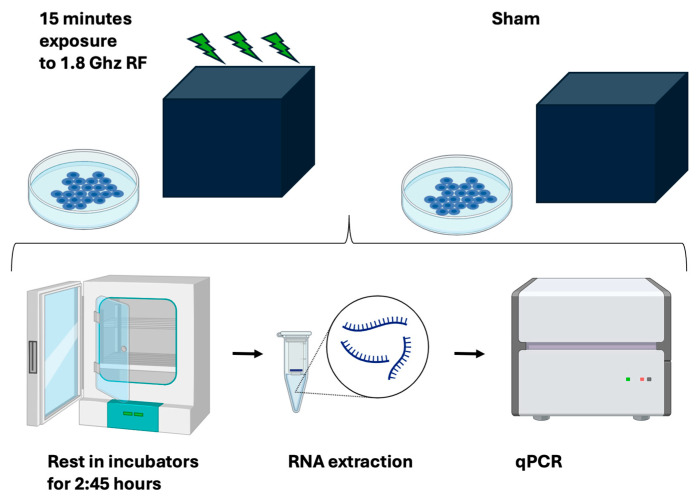
Experimental outline. Human HEK293 cells were subcultured under the identical conditions and then exposed for 15 min to either RF or to a sham signal in a custom-built RF exposure box. Subsequent to RF exposure, cell cultures were returned to their regular growth incubator for an additional 2.45 h before being frozen, undergoing RNA extraction, and then qPCR analysis to determine gene expression changes.

**Figure 5 bioengineering-12-00257-f005:**
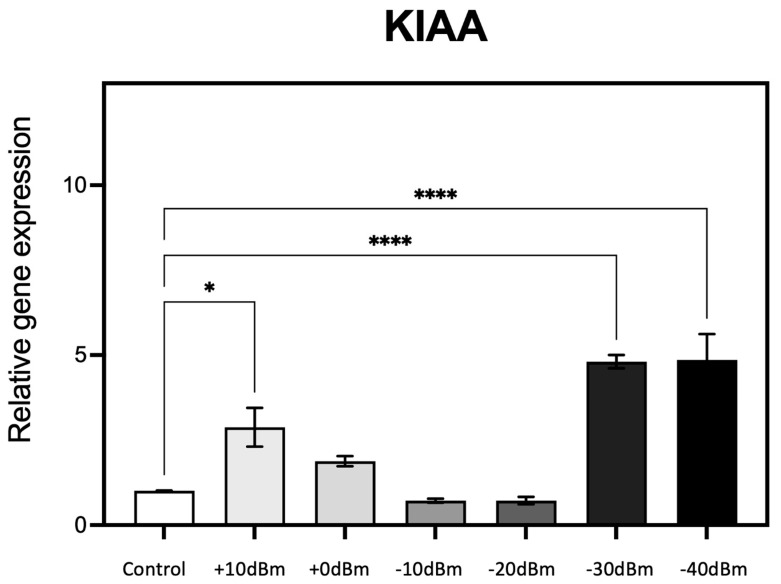
qPCR analysis of gene expression in response to the generator RF signal amplitude (see Table 2 for corresponding electromagnetic power measurements in Wm^−2^). Cell cultures grown in triplicate were exposed for 15 min to different RF signal amplitudes in the RF exposure box as described (Methods). Change in expression of the known ROS-regulated gene KIAA (20, 21) was monitored as a function of the control (Sham) exposure condition of less than P_EMG_ = −75 dBm = 3.3 × 10^−7^ Wm^−2^ (Control). *n* = 3; *p* < 0.0001 **** and *p* < 0.05 * represent *p*-value significance in comparison to sham exposure.

**Figure 6 bioengineering-12-00257-f006:**
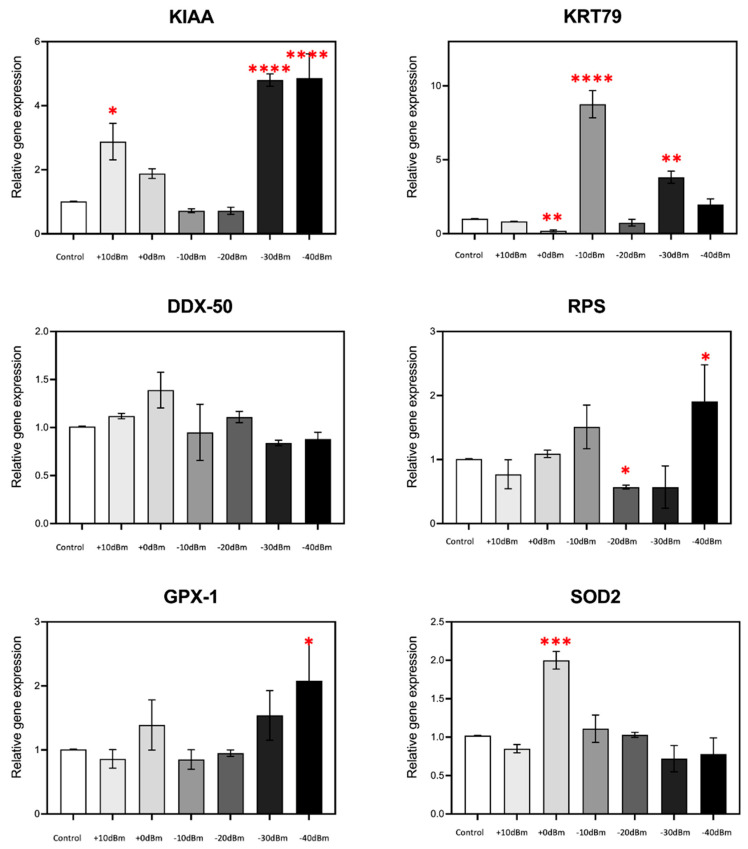
RF signal amplitude dependence of ROS-regulated gene expression is non-linear. In the present experiment, we have analyzed the expression characteristics of several genes at amplitudes ranging from control P_EMG_ = −75 dBm to P_gene_ = +10 dBm (corresponding to P_EMG_ = 2.9 × 10^−3^ Wm^−2^). Statistical significance is calculated with respect to expression levels of the control (Sham-treated) samples. *n* = 3; *p* < 0.0001 ****, *p* < 0.001 ***, *p* < 0.01 **, and *p* < 0.05 * represent *p*-value significance in comparison to sham exposure.

**Table 1 bioengineering-12-00257-t001:** Primer list.

Species	Genes	Forward Primer 5′-3′	Reverse Primer 5′-3′
Human	KIAA1211	AGCTGGCTGTTAAGCCAAAA	CCTCCAGTTCTCGCCAGTAG
	RPS16P5	TGCTAATGGCTGTGTGAAGC	GCCACAACAGGAAAAGGTGT
	KRT79	GAGGAGAGCAGGATGTCTGG	CGGTGCTATAGCCCACATTT
	DDX50	GATGTCAGCTGTGCTTGGAA	AGCCACTCCTCTGTCTGGAA
	GPX-1	TGGGCATCAGGAGAACGCCA	GGGGTCGGTCATAAGCGCGG
	SOD2	GCAGCTGCACCACAGCAAGC	CGTGCTCCCACACATCAATCCCC
	GAPDH	ATTCCACCCATGGCAAATTC	CGCTCCTGGAAGATGGTGAT

**Table 2 bioengineering-12-00257-t002:** Comparison between electrical power emitted by the generator and the electromagnetic power measured by the electric field probe.

P_gene_ [dBm]	20	10	5	0	−5	−10	−15	−20	−30	−40
P_MES_^X^ [dBm]	−29.8	−35.5	−40	−45.5	−50	−55.5	−60.3	−65.5	−72	---
E [V.m^−1^]	2.05	1.06	6.3 × 10^−1^	3.3 × 10^−1^	2 × 10^−1^	1.1 × 10^−1^	6.1 × 10^−2^	3.3 × 10^−2^	1.6 × 10^−2^	
P_EMG_^X^ [W.m^−2^]	1.1 × 10^−2^	2.9 × 10^−3^	1.1 × 10^−3^	2.9 × 10^−4^	1.1 × 10^−4^	2.9 × 10^−5^	9.8 × 10^−6^	2.9 × 10^−6^	6.6 × 10^−7^	---

## Data Availability

Data is available in the Appendix A and through request from the authors.

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
