# Peer review of "A Novel Method for Achieving Precision and Reproducibility in a 1.8 GHz Radiofrequency Exposure System That Modulates Intracellular ROS as a Function of Signal Amplitude in Human Cell Cultures"

_bioengineering, 2025, doi:10.3390/bioengineering12030257_

Round 1

Reviewer 1 Report

Comments and Suggestions for Authors

Dear Authors,

I have reviewed your manuscript titled "A Novel Method for Achieving Precision and Reproducibility in a 1.8GHz Radiofrequency Exposure System that Modulates Intracellular ROS as a Function of Signal Amplitude in Human Cell Cultures"

My General Comments: The manuscript presents an interesting and novel approach to studying the biological effects of radiofrequency (RF) exposure, focusing on intracellular reactive oxygen species (ROS) modulation in HEK293 cells. The authors have designed a custom RF exposure system and demonstrated its capability to produce reproducible gene expression results in response to RF exposure. The study's methodology, technical rigor, and relevance to public health concerns are commendable. However, several areas require clarification and improvement to enhance the manuscript's scientific quality and reproducibility.

Minor Comments:

  1. Statistical Analysis:
    • Given the observed non-linear trends in gene expression data, please justify why one-way ANOVA was preferred over non-linear regression models, which might better capture the complexity of the responses.
  2. Mechanistic Insights:
    • The manuscript provides a comprehensive discussion on ROS signaling pathways. To enhance the depth of analysis, consider elaborating on specific downstream signaling pathways involved in ROS-mediated gene expression changes.
  3. Data Interpretation:
    • The manuscript effectively describes the biphasic, non-linear, U-shaped gene expression responses. Expanding the discussion on the potential physiological implications of these patterns, particularly in the context of receptor-mediated hormetic responses, would strengthen the manuscript.
  4. Grammar and Typographical Errors: The manuscript contains several typographical and grammatical errors. For example:
    • On page 3, line 140: "The cycling conditions were as follow..." should be corrected to "The cycling conditions were as follows..."
    • On page 18, line 632: "We are indepted to Yves Chatelon..." should be corrected to "We are indebted to Yves Chatelon..."
    • A thorough proofreading is recommended to identify and correct such errors throughout the manuscript.
  1. References: Ensure all references are correctly formatted according to the journal’s guidelines. For example:
    • On page 18, reference 2: "Heating FactorHealth Phys." should be corrected to "Heating Factor. Health Phys."
    • On page 18, line 638: "Environment Inter- naional" should be corrected to "Environment Inter- national."
    • Reference 1: "https://www.sciencedirect.com/special-issue/109J1SL7CXT" is not justified. 

Author Response

Statistical Analysis:
Given the observed non-linear trends in gene expression data, please justify why one-way ANOVA was preferred over non-linear regression models, which might better capture the complexity of the responses.

Response: We appreciate the reviewer’s comment. In this paper, we used only one-way ANOVA, as described in the manuscript. Our choice is based on its ability to compare average gene expression across multiple conditions without assuming a specific pattern in the data. Although non-linear trends exist, our main goal is to test for differences between groups, not to fully model the complex behavior. Using non-linear regression would require extra assumptions and parameters that our current dataset does not support. Therefore, our analysis remains focused solely on one-way ANOVA.

Query:

  1. Mechanistic Insights:
    • The manuscript provides a comprehensive discussion on ROS signaling pathways. To enhance the depth of analysis, consider elaborating on specific downstream signaling pathways involved in ROS-mediated gene expression changes.
  2. Data Interpretation:
    • The manuscript effectively describes the biphasic, non-linear, U-shaped gene expression responses. Expanding the discussion on the potential physiological implications of these patterns, particularly in the context of receptor-mediated hormetic responses, would strengthen the manuscript

Response: We have included some extra paragraphs on ROS signaling and hormesis, in the context of our results. These are included in the modified discussion section. 

Query: 

  1. Grammar and Typographical Errors: The manuscript contains several typographical and grammatical errors. For example:

Response: We thank the reviewer for pointing out these inaccuracies and have corrected them, as well as re-proofed the manuscript.

Reviewer 2 Report

Comments and Suggestions for Authors

This paper contains important information. The data is consistent with data we have obtained at other frequencies on the growth of cancer cells and changes ROS for longer exposure times. In this reviewers opinion the key missing measurement is the lack of the static magnetic field values. Static magnetic fields in the range from 0.5 to 600 micro-Tesla have been shown to generate a similar nonlinear growth curve. Most screen rooms do shield out low frequency magnetic fields such as those generated by motors and variable power supplies. This can be done high mu shielding material such as mu-metal or coils that cancel out the external fields. It would be interesting to see what small variations in the  exposure frequencies do to the observed responses. A nonlinear response is to be expected for perturbation of natural oscillating systems in the biological material with an external signal.

of

Author Response

In this reviewers opinion the key missing measurement is the lack of the static magnetic field values. Static magnetic fields in the range from 0.5 to 600 micro-Tesla have been shown to etc.

Response:

The reviewer rightly notes that changes in the static magnetic field, or else oscillations at low frequency can also lead to cellular perturbations. However, our experimental exposure box was placed in an anechoic chamber where there was no source of external oscillation, neither machinery nor electrical equipment. There was furthermore no metal in this room, and the static field was equivalent to the local magnetic field of 50 microTessla, which did not change throughout the experiment. So, there was no detectable change (to our instruments) of any other field with the exception of the RF signal amplitude.

Conclusive proof of lack of artifact came from our biological replicates. Our exposure interval was very short (just a few minutes) and all samples were placed at the identical position for the exposure duration, before being returned to a common incubator. All experiments were performed in triplicate, providing independent biological repeats. In all of the biological repeats that we performed the differences in the replicates were not statistically significant. The only time we observed any significant variation was when we applied the RF exposure. We are therefore confident there was no artifact introduced by the experimental surroundings. This is now stated explicitly in the manuscript.